# Infection Sources and *Klebsiella pneumoniae* Antibiotic Susceptibilities in Endogenous *Klebsiella* Endophthalmitis

**DOI:** 10.3390/antibiotics11070866

**Published:** 2022-06-27

**Authors:** Kuan-Jen Chen, Yen-Po Chen, Yi-Hsing Chen, Laura Liu, Nan-Kai Wang, An-Ning Chao, Wei-Chi Wu, Yih-Shiou Hwang, Hung-Da Chou, Eugene Yu-Chuan Kang, Yen-Ting Chen, Ming-Hui Sun, Chi-Chun Lai

**Affiliations:** 1Department of Ophthalmology, Chang Gung Memorial Hospital, Taoyuan 333, Taiwan; yenpo.chen@gmail.com (Y.-P.C.); yihsing@gmail.com (Y.-H.C.); lauraliu@gmail.com (L.L.); anningchao@hotmail.com (A.-N.C.); weichi666@gmail.com (W.-C.W.); yihshiou.hwang@gmail.com (Y.-S.H.); hdmorph@gmail.com (H.-D.C.); yckang0321@gmail.com (E.Y.-C.K.); gradychen1107@gmail.com (Y.-T.C.); minghui0215@gmail.com (M.-H.S.); chichun.lai@gmail.com (C.-C.L.); 2College of Medicine, Chang Gung University, Taoyuan 333, Taiwan; 3Department of Ophthalmology, Tucheng Municipal Hospital, Tucheng, New Taipei 236, Taiwan; 4Department of Ophthalmology, Edward S. Harkness Eye Institute, Columbia University, New York, NY 10032, USA; wang.nankai@gmail.com; 5Department of Ophthalmology, Chang Gung Memorial Hospital, Keelung 204, Taiwan

**Keywords:** antibiotic susceptibility, endogenous endophthalmitis, *Klebsiella pneumoniae*, liver abscess

## Abstract

Endogenous endophthalmitis is an uncommon intraocular infection with potentially devastating consequences on vision. *Klebsiella pneumoniae* is highly prevalent in East Asian countries, with an increasing incidence recently worldwide. This retrospective study investigates infection sources and antibiotic susceptibilities of *K. pneumoniae* in patients with endogenous *K. pneumoniae* endophthalmitis (EKE) in Northern Taiwan. One hundred and fifty-seven patients with EKE were reviewed between January 1996 and April 2019. Pyogenic liver abscess (120/157, 76.4%) was the most common infection source, followed by pneumonia (13, 8.3%), urinary tract infection (7, 4.5%), and intravenous drug use (4, 2.5%). Bilateral involvement was identified in 12.1% (19/157) of patients, especially in patients with pyogenic liver abscess (16/120, 13.3%), pneumonia (2/13, 15.4%), and urinary tract infection (1/7, 14.3%). The antibiotic susceptibility rates were 98.1%, 92.5%, 97.5%, 96.8%, 100%, 99.3%, and 100% for amikacin, cefuroxime, ceftazidime, ceftriaxone, carbapenems, ciprofloxacin, and levofloxacin, respectively. Four extended-spectrum β-lactamase-producing multidrug-resistant (MDR) *K. pneumoniae* isolates were identified. In conclusion, pyogenic liver abscess was the major infection source in EKE. In addition, *K. pneumoniae* was still highly susceptible to ceftazidime and amikacin, and the MDR *K. pneumoniae* isolates were not common in EKE.

## 1. Introduction

Endogenous endophthalmitis is an uncommon intraocular infection with potentially devastating consequences on vision. *Klebsiella pneumoniae* is highly prevalent in East Asian countries [1,2,3,4], with an increasing incidence recently worldwide [5,6,7,8,9,10]. Diabetic patients with *K. pneumoniae* hepatobiliary infections have a higher risk of developing endogenous *K. pneumoniae* endophthalmitis (EKE) [1,2,3,4,5,6,7]. EKE has been found in patients with other infections, such as pneumonia and renal abscess [8,9,10,11]. *K. pneumoniae* isolates are typically susceptible to ceftazidime and amikacin, which are currently used empirically to treat endophthalmitis. Ceftriaxone is also the mainstay treatment for systemic *K. pneumoniae* infection, including liver abscess [12,13]. In the past, most studies demonstrated the antibiotic susceptibility testing results from patients admitted to the department of internal medicine. However, the details of antibiotic susceptibility testing results are lacking in patients with EKE. Amidst growing concern over the emergence of multidrug-resistant (MDR) bacteria, selecting appropriate antibiotics for infection treatment has become a critical issue [14,15]. Common variants of MDR *K. pneumoniae* include extended-spectrum β-lactamase (ESBL)-producing and carbapenemase-producing *K. pneumoniae* [14,15,16]. Although most isolates causing ocular infections are not caused by MDR bacteria, antibiotic susceptibility testing may reveal antibiotic susceptibility and resistance trends.

This retrospective study investigates the infection sources and antibiotic susceptibilities of *K. pneumoniae* in patients with EKE at a tertiary referral center in Northern Taiwan.

## 2. Methods and Materials

This study was a retrospective, consecutive case series of patients diagnosed with EKE at any time from January 1996 to April 2019 at a tertiary referral center in Northern Taiwan. The institutional review board of Chang Gung Memorial Hospital in Taoyuan, Taiwan, approved this retrospective study protocol (201900614B0C601, 10 August 2019) and waived the need for written informed consent by these patients. All clinical procedures were conducted according to the principles of the Declaration of Helsinki. The availability of medical records used either manual (1996–2000) or electronic (2001–2019) systems, and the data were collected by K.-J.C. and Y.-P.C. EKE diagnosis was confirmed if patients had signs of posterior ocular inflammation (including vitritis and chorioretinitis with or without choroidal or subretinal abscess as examined using indirect ophthalmoscopy or ultrasonography) and positive *K. pneumoniae* culture results from the blood, eye (vitreous or aqueous humor), or other extraocular sites, such as aspirated pus from an abscess, sputum, urine, or a wound. Either the primary extraocular source of infection was identified, or the presence of bacteremia was documented without a specific infection focus 

All microbiological investigations were performed at the Microbiology Department of Chang Gung Memorial Hospital, Taoyuan, Taiwan. Bacterial culture isolates were identified using conventional microbiological methods between January 1996 and December 2013 and matrix-assisted laser desorption/ionization-time of flight mass spectrometry (MALDI-TOF-MS) between January 2014 and April 2019. Conventional microbiological methods included Gram staining and biochemical tests. In MALDI-TOF-MS, automated measurement of the spectrum and comparative analysis against reference bacteria spectra were performed using an Ultraflextreme mass spectrometer and in MALDI Biotyper 3.0 software (Bruker Daltonics, Karlsruhe, Germany). The reliability of identification in the MALDI Biotyper system was expressed in terms of points. A log(score) of ≥2.0 indicated identification to the species level. The antibiotic susceptibility testing results were based on the choice of antibiotics tested in our hospital during the study period. The isolates were tested for susceptibility to various antibiotics using the Kirby–Bauer disc diffusion method on Mueller Hinton blood agar. Clinical and Laboratory Standards Institute (Wayne, PA, USA) standards were used for interpretation and quality control for each corresponding year [13]. Because we reported the visual outcomes in our previous study [2,3], data on visual outcomes were not part of this study. All statistical analyses were performed using SPSS version 23.0 (SPSS IBM Corp., New York, NY, USA).

## 3. Results

A total of 157 patients were diagnosed with EKE during the study period. The mean age of patients was 51.8 ± 13.6 years (range: 1 month to 87 years). The predominant sex was male (108/157, 68.8%). Immunocompromised status included diabetes mellitus in 105 patients (66.9%), liver cirrhosis in 10 patients (6.4%), alcoholism in 8 patients (5.1%), malignancy with immunosuppressant therapy in 8 patients (5.1%), and end-stage renal failure in 2 patients (1.3%).

### 3.1. Infectious Source

Table 1 lists the infection sources of EKE. Pyogenic liver abscess was the most common infection source, followed by pneumonia and urinary tract infection. Pyogenic liver abscess progressed with meningitis and endophthalmitis in five patients. Four users of intravenous drugs were identified with EKE. Bilateral involvement was noted in 12.1% (19/157) of patients, especially in those with pyogenic liver abscess (16/120, 13.3%), pneumonia (2/13, 15.4%), and urinary tract infection (1/7, 14.3%). However, there was no significant difference in the rate of bilateral involvement among these three groups. Table 2 shows the distribution of *K. pneumoniae* as isolated from different samples. The most common positive *K. pneumoniae* isolates were cultured from blood samples, followed by vitreous samples and aspirated fluid from abscesses.

### 3.2. Antibiotic Susceptibility Testing

Table 3 presents the antibiotic susceptibility of *K. pneumoniae* isolates; 98% of *K. pneumoniae* isolates were susceptible to ceftazidime and amikacin, which are currently used empirically to treat endophthalmitis. The tested antibiotics included penicillins, cephalosporins, aztreonam, fluoroquinolones, aminoglycosides, carbapenems, and sulfamethoxazole-trimethoprim.

Among penicillins, *K. pneumoniae* tended to be resistant to ampicillin, but more than 90% of *K. pneumoniae* isolates were susceptible to ampicillin-sulbactam, amoxicillin-clavulanate, and piperacillin-tazobactam. The susceptibility rates to aminoglycosides and sulfamethoxazole-trimethoprim were similar and more than 95%. All *K. pneumoniae* isolates had a 90–100% susceptibility rate to cephalosporins. All *K. pneumoniae* isolates were susceptible to carbapenems and levofloxacin.

### 3.3. Multidrug-Resistant Isolates

Four MDR *K. pneumoniae* isolates were identified. Table 4 shows the antibiotic susceptibility testing results for multidrug resistance. ESBL production was detected in the four *K. pneumoniae* isolates, of which only one was susceptible to amikacin. All four isolates were susceptible to carbapenems (imipenem, meropenem, or ertapenem). The four *K. pneumoniae* isolates were obtained in one user of intravenous drugs and three hospitalized patients, including one patient with splenic abscess and two with pneumonia.

## 4. Discussion

To the best of our knowledge, this is the largest case series of patients with EKE in a systemic review of the PubMed literature. In the present study, pyogenic liver abscess was the major infection source in cases of EKE, followed by pneumonia, urinary tract infection, and intravenous drug users. *K. pneumoniae* isolates were highly susceptible to ceftazidime and amikacin. In addition, the MDR *K. pneumoniae* isolates were uncommon in EKE. Most *K. pneumoniae* isolates were susceptible to third-generation cephalosporins, carbapenems, and fluoroquinolones. In immunocompromised status, diabetes mellitus was the most important risk factor, followed by liver cirrhosis, alcoholism, malignancy with immunosuppressant therapy, and end-stage renal failure. 

Because of the small number of other infection source cases, bilateral eye involvement (13% to 15%) of EKE was only identified in patients with pyogenic liver abscess, pneumonia, or urinary tract infection. There was no significant difference in bilateral eye involvement among these three groups. Six (3.8%) of the patients died during their hospital stay. The causes of death of the six patients were related to pyogenic liver abscess in four patients, pneumonia in one, and peritonitis in one. All six patients died from sepsis and multiorgan failure. The most common positive *K. pneumoniae* isolates were cultured from blood samples (79%), followed by vitreous samples (41%) and aspirated fluid from abscesses (37%). The lower positive rate of vitreous samples could result from patients treated with systemic antibiotics before EKE was diagnosed.

The increasing incidence of EKE has recently been reported worldwide, and pyogenic liver abscess is the leading cause of EKE. Pyogenic liver abscess accounted for 76.4% of the cases in our study; however, pneumonia, urinary tract infection, and intravenous drug use (IVDU) explained other, less common infection sources. IVDU-related endogenous endophthalmitis is typically caused by *Staphylococcus aureus* and *Candida albicans* [17,18,19]. There were some case reports with IVDU-related EKE [20,21]. Our findings suggest that *K. pneumoniae* should be considered a pathogen for IVDU-related endogenous endophthalmitis. Among four MDR *K. pneumoniae* strains, three isolates were hospital-acquired infections, including two patients with pneumonia and one with splenic abscess. However, the cause of one MDR *K. pneumoniae* isolate cultured from IVDU-related endophthalmitis was unknown.

Because the number of ESBL producers and AmpC β-lactamase producers has been increasing, previous studies observed a significant decrease in susceptibilities to third-generation cephalosporins and ciprofloxacin over the two decades of our study period [22]. Additionally, carbapenem-resistant *K. pneumoniae*, a new major concern emerging during the last decade because of a significantly compromised efficacy of carbapenem agents, has become a focal point of infection control [23,24]. In a case report of carbapenemase-producing *K. pneumoniae* endophthalmitis by Zhou et al. [25], the *K. pneumoniae* isolate was resistant to amikacin, ceftazidime, and carbapenems, but susceptible to polymyxin E and tigecycline. The trends of antibiotic susceptibility of *K. pneumoniae* isolates exhibited no observable change across the decades of our study period. Some sporadic ESBL-producing *K. pneumoniae* isolates were identified in one patient using intravenous drugs and three hospitalized patients, including one with splenic abscess and two with pneumonia. Moreover, no carbapenem-resistant *K. pneumoniae* isolate was identified, and four MDR ESBL-producing *K. pneumoniae* isolates were susceptible to carbapenems (imipenem, meropenem, or ertapenem) in our study.

However, the study has some limitations. First, this was a single-center retrospective case series. Second, *K. pneumoniae* isolates were not routinely tested at our hospital for susceptibility to the same antibiotics because the review was completed over the course of more than 23 years. Third, we did not retrieve accurate, detailed information on various types of endophthalmitis and visual outcomes, which may interest the readers. Finally, we did not further analyze the MDR bacteria for specific resistance genes. Nevertheless, our findings contribute to the literature in providing a thorough analysis of the antimicrobial susceptibility of *K. pneumoniae* from all patients diagnosed with EKE over 23 years at a tertiary medical center.

In conclusion, the most common infection source of detected EKE was pyogenic liver abscess. MDR bacterial endophthalmitis was not common. Amikacin and ceftazidime were observed to provide approximately equal gram-negative coverage. In MDR *K. pneumoniae* isolates, carbapenems and fluoroquinolones may serve as alternative treatments for EKE. The efficacy of multiple antibiotics against ESBL-producing or carbapenemase-producing *K. pneumoniae* will be tested in experimental and clinical endogenous *K. pneumoniae* endophthalmitis in future studies.

## Figures and Tables

**Table 1 antibiotics-11-00866-t001:** Infection sources of endogenous *Klebsiella pneumoniae* endophthalmitis.

Infectious Source	No. of Patients	Percent	Bilateral Involvement	Percent
Pyogenic liver abscess *†	120	76.4%	16	13.3%
Pneumonia ‡	13	8.3%	2	15.4%
Urinary tract infection/renal abscess	7	4.5%	1	14.3%
Intravenous drug user	4	2.5%		
Necrotizing fasciitis	1	0.6%		
Cellulitis	1	0.6%		
Brain abscess	1	0.6%		
Neck abscess	1	0.6%		
Spleen abscess	1	0.6%		
Retroperitoneal nonorganic abscess	1	0.6%		
Rectal perforation with sepsis	1	0.6%		
Peritonitis	1	0.6%		
Colon cancer with necrosis	1	0.6%		
Not identified	4	2.5%		
Total	157	100%	19	12.1%

* Three patients with hepatobiliary tract infection but no obvious liver abscess on abdominal echography or computed tomography. † Five patients with meningitis, 20 patients with pneumonia, and one patient with prostate abscess. ‡ Primary infection with pneumonia was not associated with pyogenic liver abscess.

**Table 2 antibiotics-11-00866-t002:** Distribution of *Klebsiella pneumoniae* isolated from different samples.

Sample	No. of Patients	%
Blood	124	79.0
Eye	65	41.4
Abscesses	58	36.9
Urine	30	19.1
Sputum	13	8.3
Wound swab	3	1.9
Plural effusion	3	1.9
Endotracheal aspirates	2	1.3
Ascites	1	0.6
Total	157	100

**Table 3 antibiotics-11-00866-t003:** Antibiotic susceptibility of *Klebsiella pneumoniae*.

Antibiotics	Tested	Sensitive	Intermediate	Resistant	
	Interval	No. (%)	No. (%)	No. (%)	Diseases (Number)
Ampicillin	1996–2006			75/75 (100%)	All diseases
Ampicillin-sulbactam	2007–2010	44/46 (95.7%)		2/46 (4.3%)	Liver abscess (2)
Amoxicillin-clavulanate	2010–2019	22/22 (100%)			
Ticarcillin	1996–1997			4/4 (100%)	Liver abscess (4)
Piperacillin	1996–2010	75/93 (81.7%)	5/93 (5.4%)	13/93 (14.0%)	Liver abscess (9), pneumonia (4)
Piperacillin-tazobactam	2007–2019	64/67 (95.5%)	2/67 (3.1%)	1/67 (1.5%)	Liver abscess (1)
Gentamicin	1996–2019	153/157 (97.5%)		4/157 (2.5%)	Pneumonia (2), spleen abscess (1), IVDU (1),
Amikacin	1996–2019	154/157 (98.1%)		3/157 (1.9%)	Pneumonia (2), spleen abscess (1)
Sulfamethoxazole-trimethoprim	1996–2010	91/96 (95.8%)		5/96 (4.2%)	Pneumonia (2), liver abscess (2),spleen abscess (1)
Cephalothin	1996–2000	20/20 (100%)			
Cefazolin	1996–2009	82/89 (92.1%)		7/89 (7.9%)	Liver abscess (3), pneumonia (2), spleen abscess (1), IVDU (1),
Cefuroxime	1999–2019	135/146 (92.5%)		11/146 (7.5%)	Liver abscess (7), pneumonia (2), spleen abscess (1), IVDU (1)
Ceftriaxone	1996–2019	152/157 (96.8%)		5/157 (3.2%)	Pneumonia (2), spleen abscess (1), IVDU (1), liver abscess (1)
Ceftazidime	1996–2019	153/157 (97.5%)		4/157 (2.6%)	Pneumonia (2), spleen abscess (1), IVDU (1),
Flomoxef	2007–2012	18/20 (90%)		2/20 (10%)	Pneumonia (2)
Cefepime	2008–2013	16/16 (100%)			
Cefoperazone-sulbactam	2016–2019	10/10 (100%)			
Aztreonam	1999–2009	90/95 (94.7%)		5 (5.3%)	Pneumonia (2), spleen abscess (1), IVDU (1), liver abscess (1)
Imipenem	1996–2006	82/82 (100%)			
Ertapenem	2007–2019	71/71 (100%)			
Meropenem	2007	1/1 (100%)			
Ciprofloxacin	2000–2009	130/131 (99.3%)		1/131 (0.7%)	Pneumonia (1)
Levofloxacin	2010–2019	65/65 (100%)			

IVDU, intravenous drug use.

**Table 4 antibiotics-11-00866-t004:** Antibiotic susceptibility testing in multidrug-resistant *Klebsiella pneumoniae*.

Antibiotics	*Klebsiella pneumoniae*			
No. of Isolates	1	2	3	4
Infection Source	Pneumonia	Pneumonia	Spleen Abscess	IVDU
Ampicillin	R	R	R	
Amoxicillin-clavulanic acid				S
Piperacillin	R	R	S	R
Piperacillin-tazobactam		S		S
Gentamicin	R	R	R	R
Amikacin	R	R	R	S
Cefazolin	R	R	R	R
Cefuroxime	R	R	R	R
Ceftriaxone	R	R	R	R
Ceftazidime	R	R	R	R
Floxomef	R	R	S	S
Aztreonam	R	R	R	R
Trimethoprim-sulfamethoxazole	R	R	R	
Ciprofloxacin	S	R	S	S
Imipenem	S		S	
Meropenem		S		
Ertapenem				S

IVDU = intravenous drug use; S = susceptible; R = resistant.

## Data Availability

Not applicable.

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
