# Peer review of "Infection Sources and Klebsiella pneumoniae Antibiotic Susceptibilities in Endogenous Klebsiella Endophthalmitis"

_antibiotics, 2022, doi:10.3390/antibiotics11070866_

Round 1

Reviewer 1 Report

Introduction

Please add word an appropriate word in the second sentence before 'worldwide'. 

'In the past, most studies demonstrated the antibiotic susceptibilty testing results from patients admitted in the 50 department of internal medicine' must be supported by reference.

Again 'Common variants of MDR K. pneumoniae include extended- 54 spectrum β-lactamase (ESBL)-producing and carbapenemase-producing K. pneumoniae' didn't supported by any references. Authors are requested to rewrite introduction with insertion of references.

Methods and Materials

Is it a medical record review study?, if yes then please mention it, also describe about the availability of this record as manual or electronic etc.

Who collected this data? please mention

There is confusion about the study duration, for example from 1996-2019 but bacterial identification reported to be from 2008-2013. Authors should clear it for the readers if there were conventional testing for this.

Discussion

Discussion needs to be modify and comparative studies in others health settings should be cited, self citation should be avoided overall,.

Author Response

Introduction

Please add word an appropriate word in the second sentence before 'worldwide'. 

Ans: We have rewritten the sentence.

Klebsiella pneumoniae is highly prevalent in East Asian countries [1-4], with an increasing incidence recently worldwide [5-10].     

'In the past, most studies demonstrated the antibiotic susceptibilty testing results from patients admitted in the 50 department of internal medicine' must be supported by reference.

 Ans: We add more references.

Again 'Common variants of MDR K. pneumoniae include extended- spectrum β-lactamase (ESBL)-producing and carbapenemase-producing K. pneumoniae' didn't supported by any references. Authors are requested to rewrite introduction with insertion of references.

 Ans: We add more references.

Methods and Materials

Is it a medical record review study?, if yes then please mention it, also describe about the availability of this record as manual or electronic etc.

 Ans: This is a medical record review study. We have described the availability of the medical records using either manual (1996-2000) or electronic (2001-2019) systems in the revised manuscript. 

Who collected this data? please mention

Ans: The data were collected by K.-J.C and Y.-P.C.

There is confusion about the study duration, for example from 1996-2019 but bacterial identification reported to be from 2008-2013. Authors should clear it for the readers if there were conventional testing for this.

Ans: We have rewritten this sentence. Now it reads, "Bacterial culture isolates were identified using conventional microbiological methods between January 1996 and December 2013 and matrix-assisted laser desorption/ionization-time of flight mass spectrometry (MALDI-TOF-MS) between January 2014 and April 2019."  

Discussion

Discussion needs to be modify and comparative studies in others health settings should be cited, self citation should be avoided overall,.

Ans: Thank you for your comments. We modify discussion and comparative studies in other health settings and cite more references. We have some self-citations because the case number of our previous studies was the largest in the world after the literature review.  

Reviewer 2 Report

In the manuscript ID: antibiotics-1758935, the authors propose a retrospective study about endogenous Klebsiella endophthalmitis, identifying the main infection sources and antibiotic susceptibility patterns in K. pneumoniae isolated from 157 patients at a tertiary referral center in Northern Taiwan from 1996 to 2019. The main findings confirmed pyogenic liver abscess as the principal infection source and highlighted the lack of development of significant trends of antibiotic resistance against the routinely adopted therapeutic regimens.

Although the topic is interesting, the paper does not present novel data compared to those already present in literature, which are even cited as references. Some peculiar results, as for the presence of multidrug resistant strains or the involvement of intravenous drug therapies in the development of endogenous endophthalmitis, are presents as too few cases to make relevant assumptions.

Moreover, as acknowledged by the authors, the antibiotic susceptibility profiles of K. pneumoniae isolates were not uniformly determined, a limitation that makes the results incomplete. Unfortunately, the study limitations overcome the good presented data.

The paper needs extended revisions and modifications before being considered for publication.

Author Response

In the manuscript ID: antibiotics-1758935, the authors propose a retrospective study about endogenous Klebsiella endophthalmitis, identifying the main infection sources and antibiotic susceptibility patterns in K. pneumoniae isolated from 157 patients at a tertiary referral center in Northern Taiwan from 1996 to 2019. The main findings confirmed pyogenic liver abscess as the principal infection source and highlighted the lack of development of significant trends of antibiotic resistance against the routinely adopted therapeutic regimens.

Although the topic is interesting, the paper does not present novel data compared to those already present in literature, which are even cited as references. Some peculiar results, as for the presence of multidrug resistant strains or the involvement of intravenous drug therapies in the development of endogenous endophthalmitis, are presents as too few cases to make relevant assumptions.

Moreover, as acknowledged by the authors, the antibiotic susceptibility profiles of K. pneumoniae isolates were not uniformly determined, a limitation that makes the results incomplete. Unfortunately, the study limitations overcome the good presented data.

Ans: Thank you for your comments. Because this is a long-period retrospective study, the antibiotic susceptibility testing should be different in the different years. Some old antibiotics could be omitted, whereas some new antibiotics could be tested. The testing results were based on the Clinical and Laboratory Standards Institute (CLSI) each year. In addition, most EKE is caused by pyogenic liver abscess. We highlight other infections that could be the primary source of EKE. Because most ophthalmologists and physicians may only focus on the liver abscess, the most common infection source of EKE, other infection sources may be neglected. This negligence may result in severe vision loss in patients with EKE. This is one of the purposes of our study.

The paper needs extended revisions and modifications before being considered for publication.

 Ans: Thank you for your comments. We have re-edited and revised our manuscript.

Reviewer 3 Report

The type of this article is Communication, not Original Article. This review was made with this in mind. This article significantly integrates data on rare infectious disease (endogenous Klebsiella endophthalmitis) over 20 years in one large hospital. I have a few comments:

1. I'm not sure what the second column in Table 2 means. It would be meaningful to count the number of infected eyes, but it would also be useful to count the number of patients.

2. The reference list is poor. First of all, there are too many self-citations. All of these are self-citations related to this paper, but it would be good to add other references.

3. Major risk factors for endogenous endophthalmitis are immunocompromised states (such as DM, cancer, organ transplantation) and IV drug abuse, dental procedures, etc. These are the reason for the high age of onset. There is no explanation of the patients' underlying disease - only the percentage of DM patients was mentioned. It would be good to categorize the underlying disease for each patient who had this rare infection, such as during cancer treatment or after organ transplantation.

4. minor typo error should be revised.

Author Response

The type of this article is Communication, not Original Article. This review was made with this in mind. This article significantly integrates data on rare infectious disease (endogenous Klebsiella endophthalmitis) over 20 years in one large hospital. I have a few comments:

  1. I'm not sure what the second column in Table 2 means. It would be meaningful to count the number of infected eyes, but it would also be useful to count the number of patients.

Ans: Thank you for your comments. We have changed "number of eyes" to "number of patients".

  1. The reference list is poor. First of all, there are too many self-citations. All of these are self-citations related to this paper, but it would be good to add other references.

Ans: Thank you for your comments. We add other references. We have some self-citations because the case number of our previous studies was the largest in the world after the literature review. 

  1. Major risk factors for endogenous endophthalmitis are immunocompromised states (such as DM, cancer, organ transplantation) and IV drug abuse, dental procedures, etc. These are the reason for the high age of onset. There is no explanation of the patients' underlying disease - only the percentage of DM patients was mentioned. It would be good to categorize the underlying disease for each patient who had this rare infection, such as during cancer treatment or after organ transplantation.

 Ans: Thank you for your comments. Because this is a short communication, we do not mention the details. We add other risk factors in the result section as follows, “Immunocompromised status included diabetes mellitus in 105 patients, liver cirrhosis in 10 patients, alcoholism in 8 patients, malignancy with immunosuppressant therapy in 8 patients, and end-stage renal failure in 2 patients.”

In discussion section, we add one sentence.

In immunocompromised status, diabetes mellitus was the most important risk factor, followed by liver cirrhosis, alcoholism, malignancy with immunosuppressant therapy, and end-stage renal failure.  

  1. minor typo error should be revised.

Ans: Thank you for your comments. We have re-edited and revised our manuscript.

Round 2

Reviewer 1 Report

Congratulations to the authors for the study and thank you for updating manuscript as per my comment. 

Author Response

Dear reviewer, 

Thank you for your reviewing my manuscript to improve the quality. 

Sincerely, 

Kuan-Jen Chen, MD

Reviewer 2 Report

The revised version of the manuscript ID: antibiotics-1758935 does not contain substantial differences compared to the previous one.

Regarding the previous raised criticisms:

·   --the differences in the antibiotic susceptibility testing through the years are understandable; still in the data presentation, the authors should consider distinguishing the drugs tested on all the strains, or at least in the 80% of them, from those tested on just some isolates.

·    --the main sources of infection, i.e., pyogenic liver abscess, pneumonia and urinary tract infection/renal abscess, are already described in the introduction (lines 45-48 of revised manuscript); as the other cases were identified in 1 or at maximum 4 patients in over twenty years, it is difficult to infer their relevance for the clinical practice. The authors should try to expose their data by underlining their novelty. One suggestion is to correlate the antibiotic resistance profile of the isolates with the primary source of infection, especially in the strains resulted multidrug resistant.

These changes require substantial and major modifications of the paper before being accepted for publication in “Antibiotics”.

Author Response

Dear Editor and Reviewers,

Thank you for your concerns and consideration. We try our best to revise our manuscript. We hope this revision can met the criteria of journal. The comments from the editor and reviewer are answered as follows:

The revised version of the manuscript ID: antibiotics-1758935 does not contain substantial differences compared to the previous one.

Regarding the previous raised criticisms:

  • --the differences in the antibiotic susceptibility testing through the years are understandable; still in the data presentation, the authors should consider distinguishing the drugs tested on all the strains, or at least in the 80% of them, from those tested on just some isolates.

 Ans: Thank you for your comments. We add the tested interval of antibiotics for K. pneumoniae isolates. It could be understandable for readers to know the antibiotic susceptibility testing during the study period.  

   In methods, we add one sentience as follows.

  The antibiotic susceptibility testing results were based on the choice of antibiotics tested in our hospital during the study period.

  • --the main sources of infection, i.e.,pyogenic liver abscess, pneumonia and urinary tract infection/renal abscess, are already described in the introduction (lines 45-48 of revised manuscript); as the other cases were identified in 1 or at maximum 4 patients in over twenty years, it is difficult to infer their relevance for the clinical practice. The authors should try to expose their data by underlining their novelty. One suggestion is to correlate the antibiotic resistance profile of the isolates with the primary source of infection, especially in the strains resulted multidrug resistant.

   Ans: Thank you for your comments. We add the data of antibiotic resistance profile of isolates with primary source of infection in Table 3.

    In discussion, we add some information of MDR K. pneumoniae.

Among MDR K. pneumoniae strains, three isolates were hospital-acquired infections, including two patients with pneumonia and one with splenic abscess. However, the cause of one MDR K. pneumonia isolate cultured from IVDU-related endophthalmitis was unknown. 

These changes require substantial and major modifications of the paper before being accepted for publication in “Antibiotics”.

Ans: Thank you for your comments. We do our best to edit substantial change and major modifications of our manuscript to fit this journal.

Sincerely yours,

Kan-Jen Chen, M.D.

No. 5 Fu-Hsing Street, Kuei-Shan District, Department of Ophthalmology, Chang Gung Memorial Hospital, Linkou, Taoyuan 333, Taiwan

Tel: +886-3-3281200 ext. 8671 ;  Fax: +886-3-328-7798; 

E-mail: cgr999@gmail.com

Reviewer 3 Report

The points reviewed at the beginning were corrected appropriately. I have no more suggestions or comments.

Author Response

(The authors gave the same response as above.)

Round 3

Reviewer 2 Report

In the revised version of the manuscript ID: antibiotics-1758935 the authors have improved the quality of the reported results and their interpretation. Although showing poor novel data, the manuscript meets the minimal requirements to be published in “Antibiotics” as a communication.